# OpenReview forum: "Cradle: Empowering Foundation Agents towards General Computer Control"
_ICLR.cc/2025/Conference — Submitted to ICLR 2025_

### Official Review · Reviewer_bghc · 2024-11-02

**Soundness:** 4
**Presentation:** 4
**Contribution:** 4
**Rating:** 8
**Confidence:** 3

**Summary:**

This paper presents an LLM-powered agent designed to interact with various virtual environments, processing screenshots as inputs and executing actions through mouse and keyboard controls, encapsulated within functions generated by the underlying language model. The agent is composed of several modules, including self-reflection, memory, and action planning. The authors demonstrate the agent's versatility across a wide range of environments, from complex, long-duration open-world action games to planning and decision-making games, as well as everyday software applications like web browsers and image and video editing tools.

**Strengths:**

This paper presents a well-designed framework that effectively leverages LLMs to build generalized agents capable of performing tasks across diverse domains. The agent's modular design makes it easier to extend or modify specific functionalities for different tasks and environments.

By translating language-based capabilities into embodied actions (using mouse and keyboard inputs), the paper showcases an innovative approach to applying LLMs in environments that require physical interaction without relying on a constrained, manually crafted action space.

The paper includes a comprehensive benchmark covering a variety of tasks, along with a thorough and detailed evaluation. The appendix is also highly informative, providing extensive details on every aspect of the system.

The agent's demonstrated ability to perform tasks in widely-used applications like web browsers and media-editing software suggests strong potential for real-world applications and scalability to more practical, non-gaming environments.

**Weaknesses:**

In several parts of the paper, the authors attribute the agent’s poor performance on certain tasks to the limited OCR and visual capabilities of the underlying LLM. However, no formal evaluation of these limitations is presented in the main paper. While Appendix E5 briefly discusses GPT-4's perception limitations, it lacks a systematic analysis or quantifiable assessment.

The visual perception issues are understandable in scenarios with low visibility, such as in the game RDR, but are less justified in standard software applications. When discussing CRADLE's performance on these applications, the authors cite difficulties in recognizing button states and locations due to non-standardized layouts of some tools used, stating that “lacking prior knowledge by GPT-4 leads to the failure of task inference and selecting the correct skills.” This reliance on prior layout familiarity poses a limitation to the agent’s ability to generalize, in my opinion. Ideally, the agent should identify interactive elements based on general characteristics, rather than relying on previously encountered layouts, which undermines its robustness in unfamiliar software environments.

**Questions:**

I only have a general question about how the authors propose to address the issues with impaired visual capability. Specifically, in situations like the one cited below as it seems that even if the agent can recognize every object in the scene, it would still struggle due to a lack of "attention" or focus.

> L359: “Additionally, indoor tasks like Search for Supplies are also challenging due to GPT4-o’s limited spatial perception, which finds it difficult to locate target objects and ends up circling aimlessly around the house. Moreover, the room contains numerous interactive items unrelated to the task, resulting in much more steps for the agent to complete the task.”

---

> ### Author Response · Authors · 2024-11-17
>
> We thank the reviewers for their valuable feedback and insightful comments.  We hope our following answers will clear up the doubts about our work, and please let us know if there is any other clarification we can provide.
>
> ---
>
>
> **Q1**: In several parts of the paper, the authors attribute the agent’s poor performance on certain tasks to the limited OCR and visual capabilities of the underlying LLM. However, no formal evaluation of these limitations is presented in the main paper. While Appendix E5 briefly discusses GPT-4's perception limitations, it lacks a systematic analysis or quantifiable assessment.
>
> **A1**: Thanks for pointing it out. Due to the limited pages in the main paper, we mainly focus on introducing GCC setting and Cradle framework and try our best to provide additional useful information in the appendix. Systematic analysis of the LMMs visual capabilities is indeed important, which is worth a separate paper for study. There are already some recent works[1] on it, whose conclusion is similar to ours that "VLMs consistently struggle with tasks that require precise spatial information and recognizing geometric primitives that overlap or are close together."
>
> [1] Rahmanzadehgervi, Pooyan, et al. "Vision language models are blind." ACCV 2024.
>
> ---
>
> **Q2**: The reliance on prior layout familiarity poses a limitation to the agent’s ability to generalize, in my opinion. Ideally, the agent should identify interactive elements based on general characteristics, rather than relying on previously encountered layouts, which undermines its robustness in unfamiliar software environments.
>
> **A2**: Thanks for the insightful comment, which is actually not contradictory to our statement in the paper. The reliance on prior layout familiarity is exactly caused by the fact that current LMMs do not have the ability to reliably identify the interactive elements based on general characteristics. As we mentioned in the paper, take Outlook as an example, when we are writing an email, we need to click on the blank area to the right of the "To" element and then enter the email address, rather than clicking on the "To" itself. We find that it is quite difficult for GPT-4o to enter the address successfully since the blank area does not look like an interactable element. Instead, it will keep trying to click the "To" element, which in their vision looks more like an interactable button. It is a common challenge for all web/software agents and there is still significant room for improvement in LMM's UI understanding.
>
> ---
>
> **Q3**:  How do the authors propose to address the issues with impaired visual capability. In situations like the one cited below as it seems that even if the agent can recognize every object in the scene, it would still struggle due to a lack of "attention" or focus.
>
> L359: "Additionally, indoor tasks like Search for Supplies are also challenging due to GPT4-o’s limited spatial perception, which finds it difficult to locate target objects and ends up circling aimlessly around the house. Moreover, the room contains numerous interactive items unrelated to the task, resulting in much more steps for the agent to complete the task."
>
>
> **A3**: Recognizing objects in the wild is much more challenging than recognizing objects independently. The lack of "attention" or focus mentioned by the reviewers can still be explained by the fact that the model is not robust and generalized enough, where the performance has a significant drop in complex cases. The general solution is always improving LMMs’ visual understanding abilities, however, it may not be affordable for most researchers. Another temporal solution is to apply advanced domain-specific tools to augment the visual ability, like Grounding Dino and SAM. These tools are usually more lightwise and have better performance in their own domain compared to LMMs.

---

> > ### Comment · Reviewer_bghc · 2024-11-19
> > **Thanks for your reply**
> >
> > Thank you for addressing my questions. Regarding Q3:
> >
> > > Another temporal solution is to apply advanced domain-specific tools to augment the visual ability, like Grounding Dino and SAM. These tools are usually more lightwise and have better performance in their own domain compared to LMMs.
> >
> > It seems you leveraged these tools in this paper but still faced issues with image understanding.

---

> ### Author Response · Authors · 2024-11-19
>
> Thanks for pointing it out. Yes, that is why we call it a temporal solution. Though it cannot totally solve the problem, it indeed alleviates the issue. We also want to point out that these tools have improved quickly. For example, when we were developing Cradle, what we used were Grounding Dino and SAM. In recent days, Grounding Dino 1.5 and SAM 2 are already available. A performance gain would be expected if these tools are replaced with the latest advanced version.

---

### Official Review · Reviewer_uF4M · 2024-11-04

**Soundness:** 3
**Presentation:** 3
**Contribution:** 3
**Rating:** 8
**Confidence:** 3

**Summary:**

The paper introduces CRADLE, a modular framework utilizing large multimodal models (LMMs) for general computer control (GCC). The authors evaluate its capabilities on both video games and software applications, illustrating the system's adaptability across various domains. This presents an interesting contribution to the conference and offers valuable insights for advancing agent-based control systems.
The framework is composed of six distinct modules: information gathering, self-reflection, task inference, skill curation, action planning, and memory.

**Strengths:**

This framework demonstrates considerable complexity, with extensive evaluations across video games and diverse software applications. The use of a standard action space is particularly effective, allowing the framework to generalize across applications with minimal adaptation.

The system's performance is noteworthy; however, it's somewhat unclear to what extent this success is attributable to the baseline human players' lack of expertise in the evaluated games, as they had limited practice compared to the agent, which was afforded many trial steps.

The ablation study, examining each module’s contribution, provides valuable insights.

**Weaknesses:**

While the evaluations are extensive, the number of runs and human participants remains relatively small to achieve robust statistical significance. Given the complexity and resources required for such evaluations, the authors might consider acknowledging this limitation explicitly in the paper.

**Questions:**

Line 1197: Could the authors clarify the specific visual prompting techniques employed, particularly the extent of manual curation involved? Greater transparency here would help assess the generalizability of the framework.

Were task-specific prompts manually written, or were they generated—at least partially—by LMMs? If generated, what iterative methods were employed to refine these prompts?

Can the authors elaborate on the design rationale for each module, discussing why each module was considered necessary and the factors underlying their inclusion, rather than directly presenting each module?

For novel tasks, how much time and expertise are required to develop prompts and input videos for CRADLE to generalize effectively? An exploration of these requirements could provide a clearer picture of the framework's adaptability.

---

> ### Author Response · Authors · 2024-11-17
>
> We thank the reviewers for their valuable feedback and insightful comments.  We hope our following answers will clear up the doubts about our work, and please let us know if there is any other clarification we can provide.
>
> ---
>
> **Q1**: While the evaluations are extensive, the number of runs and human participants remains relatively small to achieve robust statistical significance. Given the complexity and resources required for such evaluations, the authors might consider acknowledging this limitation explicitly in the paper.
>
> **A1**: Thanks for pointing it out and we deeply appreciate the reviewer's understanding of the complexity of the evaluations. We totally agree that the limited number of human participants is one limitation of this paper. We have updated it in the latest version.
>
> ---
>
> **Q2**: Could the authors clarify the specific visual prompting techniques employed, particularly the extent of manual curation involved? Greater transparency here would help assess the generalizability of the framework.
>
> **A2**:  Thanks for pointing it out. Due to the limitation of pages, the visual prompting techniques we used were introduced in Figure 21 (Page 36) and Figure 29 (Page 46) in the appendix. The main idea of them is to use colorful bands to improve LMMs’ sense of direction and draw coordinates and axis on the screenshots to improve LLMs’ visual understanding and pixel-level control.
>
> ---
>
> **Q3**: Were task-specific prompts manually written, or were they generated—at least partially—by LMMs? If generated, what iterative methods were employed to refine these prompts?
>
> **A3**: When we start to work on a new software/game, we will first ask LMMs to analyze task/domain-specific stuff that each module needs to pay attention to, which forms the initialization of the domain knowledge into the prompt template. Then we will iteratively modify the prompts according to the agent's performance manually with the help of LMMs, to make sure the prompts align with LMMs conventions and are free of ambiguity. How to let LMMs generate the prompts automatically and iteratively is an insteresing and promising open question.
>
> ---
>
> **Q4**: Can the authors elaborate on the design rationale for each module, discussing why each module was considered necessary and the factors underlying their inclusion, rather than directly presenting each module?
>
> **A4**: As we introduced in the paper, the philosophy behind the design of the modules is a straightforward problem-solving mindset, whose reasoning process is analogous to "reflect on the past, summarize the present, and plan for the future".
>
> Information Gathering and Self-Reflection are used to analyze the performance and outcomes of the preceding action's execution.
>
> Task Inference and Memory summarize the current situation and analyze the current progress of the task based on the reflection results.
>
> Skill Curation and Action Planning generate and select skills to be executed in the next actions,
> according to the current situation and the progress of the task.
>
> We are glad to see the framework motivated by the intuitive but powerful problem-solving mindset works well across domains.
>
> ---
>
> **Q5**: For novel tasks, how much time and expertise are required to develop prompts and input videos for CRADLE to generalize effectively? An exploration of these requirements could provide a clearer picture of the framework's adaptability.
>
> **A5**: One example we can provide is that a group of amateurs managed to develop Cradle into the
> Black Myth: Wukong and defeated some Bosses within 2 weeks. Note that this game is an extremely challenging action game and it was released 3 months ago, which is later than the knowledge cutoff date of GPT-4o.

---

> > ### Comment · Reviewer_uF4M · 2024-11-26
> > **Thanks for the reply**
> >
> > A4:
> >
> > "is a straightforward problem-solving mindset" -- There are probably a lot of different framework of problem solving. I would suggest tuning the language down, and find any reference to cognitive science that supports the problem solving framework you are implicitly using. Or acknowledge this is a framework the team came up with, but other frameworks could exist and can be explored grounded on cognitive science in the future.
> >
> > A5:
> >
> > Would be good to add to the paper as clarification.

---

> > > ### Author Response · Authors · 2024-11-26
> > >
> > > **A4:**
> > > Thanks for the insightful comments. The framework is indeed proposed by our team. As we mentioned several times in the paper, Cradle is the preliminary/first attempt towards GCC setting to show the effectiveness of GCC, which is definitely not the ultimate answer or the only possible framework. We have added this to the Limitation and Future Work Section in the latest version of the paper that exploring other potential frameworks under GCC setting is also a promising future direction. It is also the intention of Cradle Project to attract more researchers to join this exciting and challenging domain.
> > >
> > > **A5:**
> > > Thanks for the constructive suggestion. Due to the double-blind policy, it is a pity that we cannot mention this work in the submission. In the camera-ready version, we will add it to our paper.

---

### Official Review · Reviewer_pm9Q · 2024-11-04

**Soundness:** 3
**Presentation:** 1
**Contribution:** 2
**Rating:** 5
**Confidence:** 3

**Summary:**

The paper presents an LLM powered system/framework for solving tasks by direct observation of screen state (i.e. from pixels) and direct manipulation of computer controls such as  mouse and keyboard. The system comprises of 6 high level modules and the authors demonstrate performance on a range of difficult tasks including open-world game playing and UI navigation on a benchmark of typical computer use/productivity tasks. The novelty lies in the development of a particular framework to use pretrained VLM based controller (as opposed to a model learned from scratch) to perform control from pixels (as opposed to a tailored representation of the state of the world).


Note:
I've updated score after the author rebuttal/clarification.

**Strengths:**

I do applaud the immense amount of effort that has been put into this work and a big strength of this work is the evaluation across a wide variety of task types that include game and non-game settings. The results presented also seem to be relatively strong. The primary contribution is an exploration of what ingredients are needed to get an LMM like GPT-4o to complete a range of tasks when limited to input via pixels and output via keyboard and mouse actions (vs api's provided by these software) in a similar manner to the way a human would control these systems. The system also has to some degree infer what actions it can take in the world and how to do so which is also challenging to do.

**Weaknesses:**

I think my primary concerns are around the presentation of the work, overall I found it quite hard to extract takeaways about the design of the system and to understand the experiment results from the *main body of the paper*. This is partly because of the scope of the work and the evaluation scenarios, but I think more could be done to bring the main arguments and results into the body of the paper rather than in the appendices (and the appendices are quite long). Overall I'd encourage the authors to consider more what are they key things they would want a reader to takeaway from the paper (and potentially apply to future systems).

Some specific examples of the challenges I had:
- Its quite hard to understand the design/implementation of the 6 modules in Cradle. Aside from high level descriptions of the *purpose* of the module little information is presented about how they function in the main text. I think parts of the *general implementation* that is currently in the appendix need to move to the main body.
- It is hard to understand the critical differences and similarities between this framework and that used in other systems including ones cited such as, Describe, Explain, Plan and Select, Jarvis-1, etc. Is it the addition of external tools to prompt VLM perception of the scene, or a particular component that you would say is the main advancement in the design of such systems.
- Clearer discussion of the bootstrapping ) the system requires to get started in the various scenarios would be helpful to discuss upfront (e.g. initial skills, and domain information provided in the prompts for each component). It seems like there is quite a bit of upfront work to get Cradle working in a new domain. These are listed in one of the appendices, but some guiding principles/base requirements would be useful for readers to know.  Quantitatively the authors could for example present the number of low-level and composite skills provided for each scenario and the number of generated skills to complete the task successfully.
- A clearer presentation of the quantitative task performance before diving into qualitative analysis of the many different behaviors would make it easier to follow:
	- The paper focuses a fair amount on the results from RDR2, but could better contextualize the tasks themselves (e.g. their difficulty) and the behavior of the system when performing them e.g. how many different discrete actions need to be executed to complete each task—does following an NPC only require executing a single *follow* action (that is my impression after reading the appendix).
	- What is the success rate of the system on RDR2 out of the 5 runs? Do they all compete successfully?
- The baseline comparisons in section 4.1. do not seem all that useful/informative to me. It is not clear what is being 'ablated' compared to Cradle (GPT-4o) in Table 3. While i think the goal was to try and benchmark in some way against an existing system. I think it would be better to focus on ablations of the different parts of the system itself such as in 4.2. My understanding from reading this section is that the main difference in the Reflextion-like is the lack of "Task Inference" stage, but looking at the ablations in the next section, ablating Task Inference had relatively low impact compared to the other components? Am i understanding this correctly?
- The ablations in 4.2 could be expanded to at least one non-game scenario tasks. The authors present the system as one that **generalizes across multiple domains** and it would be useful for readers to know what elements of the system are most helpful for enabling that generalization or indeed if non-video game domains need the same set of components as video game domains. I'd encourage the authors to run the ablations over the OSWorld benchmark or a benchmark like WebArena (which has wider adoption) that enables automatic evaluation
	- Another ablation that I think would make this a stronger paper is a version of Cradle where the LLM doesn't have access to tools such as Grounding DINO, SAM and others. Is that they key ingredient to make systems like this effective.
- Some of the figures and tables are hard to understand
	- Table 1: What are the units for *"Farm Clearup (Grids Num)"*  and *"Avg Haggling (Count)"*?
	- Table 1: When you use the plus or minus sign, is that reflecting the range in both directions? Cities Skylines Human population is 415+-416, which would imply some player had -1 population? This table might be clearer if it indicated the min and max separately.
	- Table 4: I think this should be be "Figure" 4. I do find it hard to read, what do the shaded areas represent? They seem to overlap a fair amount, how should we interpret this?
	- Figure 6 is quite busy, the annotations look like the data points being plotted, I don't know if the chart is helped by the connecting lines. It might be more helpful to see average number (and std-dev) of steps per task rather than cumulative steps to allow readers to better understand performance on hard vs difficult tasks. The top half of the figure (the two maps) isn't all that helpful and could give you back space to bring in more information about the system.

I am open to increasing my score, particularly if the authors can help articulate what the main things they think the community should learn/takeaway from this paper (other than the existence of the system)

**Smaller nitpicks (does not affect the score)**

L094: "We managed to prove that commercial software is out-of-box testbeds under our framework." — Not very clear what this sentence means.

**Questions:**

Are the authors proposing the GCC (General Computer Control) setting as a contribution/novel aspect of this work? Could the authors clarify how the proposed GCC differs from the general approach to learning control from pixels *Human-level control through deep reinforcement learning* [Mnih et al 2016]?
- How is a 'step' defined in the context of the games/cradle? Is one step=one action selected by cradle? How many mouse/keyboard interactions can happen in one step?
- In table 1 are the humans limited in the amount of time they get to spend to complete the tasks?
- How customized do the prompts for each module in Cradle need to be for each domain it is applied to? Is there anything that can be shared across multiple domain without heavy customization?
- What happens to visual information/screenshots during the summarization process into episodic memory?
- In appendix H.4 how is "visual prompting" different from the information gathering used in other scenarios? If the augmented screenshot is not fe d to the information gathering modules, where does it go?

---

> ### Author Response · Authors · 2024-11-17
>
> We thank the reviewers for their valuable feedback and insightful comments.  We hope our following answers will clear up the doubts about our work, and please let us know if there is any other clarification we can provide.
>
> ---
>
> **Q1**: About the presentation of the work.
>
> **A1**: We deeply understand the reviewer’s confusion from the comments and sincerely appreciate the reviewers’ efforts and time to write such a long review. However, we are quite confident that the paper tells a complete story. The takeaways are apparent and easy to get, which are clearly stated in the abstract, introduction and conclusion sections.
>
> After carefully reading the reviewers’ review more than ten times, we believe the main conflict comes from the different positions and expectations, namely the contributions that we have regarding Cradle framework in this paper, rather than the presentation, so that the reviewers may expect more presentation which is actually not the focus of the paper. It is partially caused by the reason that the reviewers are not familiar with the domain and the related works, resulting in difficulties in understanding the General Computer Control (GCC) setting. We here try our best to make some clarifications to bridge the domain gap.
>
> **1.1 GCC Setting**
>
> The primary contribution of this paper should be attributed to the GCC setting. The GCC setting provides a blueprint for how the agents can interact with any software without access to the built-in APIs, whereas almost all of the previous works heavily rely on the APIs resulting in the lack of generalizability.
>
> This setting mainly has three advantages:
> 1. It could save huge efforts in designing specific wrappers for computer environments. Traditionally, researchers have to design specific observation spaces and action spaces to encapsulate the raw environments into research environments, which usually needs a comprehensive understanding of the environment and a period of hard work. Agents under GCC settings can directly be deployed into any computer environment in the wild with the unified interface.
>
> 2. It greatly extends agents’ reach to the tasks and software that have never been explored due to the lack of existing research encapsulation. Most current research environments are based on open-source environments, where they can directly obtain the internal state information and execute various actions through APIs with the designed encapsulation. The encapsulation is difficult to design in close-sourced software without access to built-in APIs. Under our GCC setting, agents can directly interact with software in the wild without the precondition of access to APIs. They can be developed into enormous commercial software, especially video games, which are precious undiscovered treasures, to learn and evaluate various abilities.
>
> 3. Under GCC setting, all the data is collected in the same format and granularity since agents interact with various environments in a unified way, which enables comprehensive training for general improvement towards general intelligence.

---

> > ### Author Response · Authors · 2024-11-17
> >
> > **1.2 Cradle Framework**
> >
> > The second contribution then comes to the open-sourced Cradle framework. As the **first attempt** towards GCC, Cradle is used to prove that the challenges that GCC presents (which previous works tend to avoid) can already be properly handled by the current techniques. Cradle not only provides a feasible solution to the most essential GCC challenge of how to process raw input from the monitors and control keyboard and mouse but also explores how to improve the performance on challenging long-horizontal computer tasks.
> >
> > In Section 3.1 Environment IO, we provide a detailed introduction to how Cradle deals with screenshot/video input and output keyboard and mouse actions, which is a unique challenge in the GCC setting. The other modules try to improve the performance on challenging long-horizontal computer tasks, which is actually a common challenge for all the computer agents no matter whether they are under GCC setting. We only provide a high-level description for some modules, like episodic memory, procedural memory, self-reflection and so on, since these modules are kind of standard components in LLM-based agents. The functions of these modules are straightforward and can be found in many previous famous works, like Reflextion[1] and Voyager[2]. Just like in an LLM paper, we do not need to waste too much space to explain how the transformer blocks work if without any modification. We did not claim any contribution from any independent module, instead, the novelty comes from the whole Cradle framework as an integration.
> >
> > We also do not think it is proper to move the details in the general implementation section in the appendix to the main paper, which will harm Cradles’ generalizability and mislead the readers. Cradle is a general framework instead of a specific method, where each module can have different env-specific implementations to deal with env-specific issues. The general implementation in the appendix introduces how we empirically implement Cradle at the code level. It is limited to the available tools and LMMs when developing, which updates very fast these days. And the way we implement the modules could always be improved. It is unnecessary to mention these temporary technical details in the main paper. And we also agree that the technical details are very important, so we tried our best to provide all the details we can provide in the appendix for the readers and open-source our code to the community.
> >
> > Besides, we believe Figure 3 and 5 are sufficient to demonstrate the input & output and provide a precise understanding of our framework. At the beginning of the framework section, we use Figure 3 to show the input and output of each module to provide a high-level understanding to the readers. Then in Figure 5, we provide 3 complete examples in different games to show how Cradle works during each step with real input and output examples.
> >
> >
> > **1.3 Performance and Impact**
> >
> > The third and final contribution is that Cradle’s superior performance on 4 challenging video games, 5 software applications and a comprehensive benchmark, OSWorld, manages to show its effectiveness and the feasibility of GCC setting. Even the extremely challenging AAA games are now within the reach of LMM-based agents and can serve as comprehensive testbeds. It also shows that there is still much space for improvement, especially in the LMMs visual abilities, pointing out the directions for future research. We hope our paper will motivate more researchers to build their own GCC agents and deploy them to their desired software environments based on our open-sourced framework. Due to the double-blind policy, we cannot provide exact links. We are delighted to see that Cradle has been developed into many interesting and challenging games by researchers and amateurs, even in the recent most popular game, Black Myth: Wukong.
> >
> > **1.4 Takeaways**
> >
> > After introducing our contribution in detail, we summarize the takeaways in brief.
> >
> > 1. **Computer agents should more consider our GCC setting**, instead of relying on built-in APIs. Our GCC setting is applicable to all the video games and other software applications, which is the future of computer agents. The latest computer use feature proposed by Anthropic is also under GCC setting.
> >
> > 2. **Cradle framework proves that the challenging GCC setting can already be properly handed by the current techniques, where both video games and other software applications can be controlled by one framework**. Meanwhile, there is still significant space for further improvement. We hope our work can motivate more researchers to join this promising direction.
> >
> > 3. **With our open-sourced Cradle framework, researchers and amateurs can easily build their own GCC agents and deploy them to their desired software environments for further research and development.**
> >
> >
> > We hope the above clarification can help reviewers better understand this work and re-evaluate the paper.

---

> > > ### Author Response · Authors · 2024-11-17
> > >
> > > **Q2**: It's quite hard to understand the design/implementation of the 6 modules in Cradle. Aside from high level descriptions of the purpose of the module little information is presented about how they function in the main text. I think parts of the general implementation that is currently in the appendix need to move to the main body.
> > >
> > > **A2**: Please refer to Q1 1.2.
> > >
> > > ---
> > >
> > > **Q3**: It is hard to understand the critical differences and similarities between this framework and that used in other systems including ones cited such as, Describe, Explain, Plan and Select, Jarvis-1, etc. Is it the addition of external tools to prompt VLM perception of the scene, or a particular component that you would say is the main advancement in the design of such systems.
> > >
> > > **A3**: As we state clearly in the related work, the critical difference is that these works do not obey GCC setting. For example, since Minecraft is an open-sourced game where users can obtain the internal
> > > state information (ground truth), and execute actions by calling game-provided APIs. These methods are difficult to transfer to the other domain, where the APIs are provided in different ways or even the APIs are not provided. The main advancement of Cradle comes from the whole framework so that it can properly solve the challenges that GCC presents rather than any independent module.
> > >
> > > ---
> > >
> > > **Q4**: Clearer discussion of the bootstrapping ) the system requires to get started in the various scenarios would be helpful to discuss upfront (e.g. initial skills, and domain information provided in the prompts for each component). It seems like there is quite a bit of upfront work to get Cradle working in a new domain. These are listed in one of the appendices, but some guiding principles/base requirements would be useful for readers to know. Quantitatively the authors could for example present the number of low-level and composite skills provided for each scenario and the number of generated skills to complete the task successfully.
> > >
> > > **A4**: We deeply understand the reviewers’ comments, however, we sincerely hope that the reviewers can take the limited pages into account. Given the limited space of the paper, we always need to put the context with the highest priority in the main paper. Just like hyperparameter tuning, prompt design is one of the unavoidable standard processes in LLM-based agents. We do not think every paper needs to provide principles for hyperparameter tuning in the main content, as long as they provide the hyperparameters they use in the appendix. And we do not think cutting-edge papers have to remind readers that when applying the network to a new domain, they need to tune the hyperparameters again. In the README of our provided repo, we indeed provide a detailed tutorial to show how to apply the framework to a new domain hand-by-hand at the code level.
> > >
> > > The initial skills tell the same story. We have 4 games, 5 software applications, and a benchmark. It is not possible to introduce the skills they use one by one in the main paper. Only providing the number of skills without further introduction will only cause more confusion in the main paper. We believe that providing a detailed introduction to them in the appendix would be a better solution. These technical details do not have to appear in the main paper where the space is limited and valuable.
> > >
> > > ---
> > >
> > > **Q5**: The paper focuses a fair amount on the results from RDR2, but could better contextualize the tasks themselves (e.g. their difficulty) and the behavior of the system when performing them e.g. how many different discrete actions need to be executed to complete each task—does following an NPC only require executing a single follow action (that is my impression after reading the appendix).
> > >
> > > **A5**: Due to the limited space in the main paper, we provided a detailed introduction including the difficulty in Figure 8 in the appendix. The number of discrete actions for each task is also provided in Figure 6. Tasks like Follow Dutch and Follow Javier only mean that the main activity of this task is following. During the task, the agent also needs to interact with different NPCs and items.
> > >
> > > ---
> > >
> > > **Q6** What is the success rate of the system on RDR2 out of the 5 runs? Do they all compete successfully?
> > >
> > > **A6**: Yes. As we mentioned in the paper, if a task fails, Cradle can select the ’retry checkpoint’ option (a mechanism provided by the game) to retry the task until it completes and moves to the next task. So do the human players.

---

> > > ### Comment · Reviewer_pm9Q · 2024-11-23
> > >
> > > Thanks for your details response, this section and the one above give me a better sense of the setting you are contextualizing your work within and the contributions you focus on. As I understand it, the distinction of the GCC setting is the _combination_ of: 1) the use of language model driven agents, 2) input from pixels and action space of keyboard mouse actions (as opposed to any internal APIs),  3) absence of extrinsic/custom-designed reward signal,  3) generality of application to many different settings.
> > >
> > > To your point about the original score reflecting different positions and expectations wrt this paper, I suspect you are right. With respect to Cradle as a framework, a lens I am taking is what does the framework provide to me as a potential user vs what do I need to bring to the framework to adapt it to new scenarios (and how does this framework differ from other frameworks). This is where I likely expected more description of some details of the framework compared to what you wanted focus on. Or what elements I need to customize vs ones the framework automates. Thank-you for your clarification here that the modules themselves are aren't the contribution you are making but rather putting these existing pieces together. I don't think these components are quite as standard as transformer blocks... but again, that's just my perspective.
> > >
> > > I'll update my score to reflect my current understanding based on your clarifications.

---

> > > > ### Author Response · Authors · 2024-11-24
> > > >
> > > > We sincerely appreciate the reviewer's recognition of our position wrt this paper. We are also willing to seriously consider the reviewers' valuable feedback that our paper needs to place greater emphasis on the reading experience of potential users.
> > > >
> > > > In the latest paper, we added a new subsection, **4.1 General Implementations** on page 6. This section attempts to provide more details and guidance on Cradle's practical implementation, including input, skills, prompts and how to deploy Cradle to new software, which are the key issues that potential users are most concerned about. Due to the extremely limited space in the main paper, this subsection can only include the most important points regarding implementations, please refer to the appendix for more details. We hope it will provide a clearer overview of how Cradle works in practice.
> > > >
> > > > Here is the copy of section 4.1 from the latest version of the paper:
> > > >
> > > > **4.1 General Implementations**
> > > >
> > > > **Input:** Cradle applies gpt-4o-2024-05-13 as backbone. It only takes a video clip, which records the execution progress of the last action, as input. To lower the frequency of interaction with backbone models and reduce the strain on the computer, video is recorded at 2 fps, which proves to be sufficient in most cases for information gathering without missing any important information.
> > > >
> > > > **Skills:** Cradle uses Python code to simulate keyboard and mouse operations, which is encapsulated by an io_env class to achieve OS-agnostic interface.
> > > > Skills are generated based on these basic operations. We use OpenAI's text-embedding-ada-002 model to generate embeddings for each skill, stored in the procedural memory and retrieved according to the similarities.
> > > >
> > > > **Prompts:** Prompts used by each module are initialized by the corresponding templates in Markdown-style format. These prompt templates provide a minimal workflow with basic rules for the module to run and use placeholders of each key for input and output. Cradle automatically retrieves the corresponding value for each key in the input from the episodic memory and forms valid requests to query LMMs with the values and templates. After receiving responses from LMMs, Cradle automatically extracts the keys in the output and stores them in the episodic memory. Users can freely customize their own prompts without writing any code.
> > > >
> > > >
> > > > **Apply to new environments.** Theoretically, Cradle can be directly deployed to new video games or other software applications with the default prompt templates and empty procedural memory. Due to the limited ability of current LMMs and the complexity of challenging environments and tasks, prompt engineering may need to be applied to every module to enhance LMMs' reasoning ability and introduce domain knowledge. Additional tools can also be applied to provide extra grounding and domain knowledge as part of the prompt input. Procedural memory can be initialized with hand-craft skills to mitigate the incomplete tutorials provided by the software and the complexity of tasks. Users may need to analyze the task-specific issue and choose a suitable solution. We provide all the implementation details and prompts we use for each software in Appendices D to K.

---

> > > > ### Comment · Reviewer_uF4M · 2024-11-26
> > > > **+1**
> > > >
> > > > "I suspect you are right. With respect to Cradle as a framework, a lens I am taking is what does the framework provide to me as a potential user vs what do I need to bring to the framework to adapt it to new scenarios (and how does this framework differ from other frameworks). This is where I likely expected more description of some details of the framework compared to what you wanted focus on. Or what elements I need to customize vs ones the framework automates. Thank-you for your clarification here that the modules themselves are aren't the contribution you are making but rather putting these existing pieces together. I don't think these components are quite as standard as transformer blocks..."
> > > >
> > > > I agree with this point completely, that's why I asked Q4 in my review.
> > > > It's one possible framework and design, but it's not as straightforward or standard.
> > > > Would be great to clarify that other designs are possible.

---

> > > > > ### Author Response · Authors · 2024-11-26
> > > > >
> > > > > Thanks for the insightful comments. As we mentioned several times in the paper, Cradle is the preliminary/first attempt towards GCC setting to show the effectiveness of GCC, which is definitely not the ultimate answer or the only possible framework. We have added this to the Limitation and Future Work Section in the latest version of the paper that exploring other potential frameworks under GCC setting is also a promising future direction. It is also the intention of Cradle Project to attract more researchers to join this exciting and challenging domain.
> > > > >
> > > > > As the deadline for paper revision is approaching, we would like to know whether there are other clarifications we can make to clear up the reviewers’ concerns. We deeply appreciate the different perspectives that the reviewers provide, which greatly improves the readability of our paper to various readers.

---

> > > > > > ### Author Response · Authors · 2024-11-28
> > > > > >
> > > > > > Dear Reviewer pm9Q,
> > > > > >
> > > > > > We sincerely appreciate your constructive suggestions and engagement during the rebuttal process, which greatly helps us improve the quality of our paper to more potential users. As the rebuttal deadline approaches, we would like to know whether there are any additional concerns we can address. We will try our best to provide further clarification.
> > > > > >
> > > > > > GCC is a promising but extremely challenging setting where most current works try to avoid it resulting in limited generalizability. **Cradle manages to show the effectiveness of GCC setting, which we hope can attract more researchers to engage in this exciting domain to achieve the general computer agents. And Cradle also serves as the first open-sourced framework for the researchers to start with. None of the above-mentioned will be possible without your support.**
> > > > > >
> > > > > > We hope our provided response during the rebuttal addresses the reviewers' concerns and the reviewers can re-evaluate our paper.  We thank the reviewers once again for their efforts.
> > > > > >
> > > > > >
> > > > > > Best regards,
> > > > > > Authors of Paper 6747

---

> > > > > > > ### Author Response · Authors · 2024-12-02
> > > > > > >
> > > > > > > Dear Reviewer pm9Q,
> > > > > > >
> > > > > > > We want to again express our gratitude to your efforts and insightful comments during the rebuttal period. As it is the **last day** of rebuttal for reviewers to reply, we are eager to know whether our
> > > > > > > latest revision of the paper manages to solve your remaining concerns. We added a new subsection 4.1 General Implementations on page 6 to provide more details and guidance on Cradle's practical implementation. This subsection provides a **clearer overview of how Cradle works in practice to potential users**.
> > > > > > >
> > > > > > > GCC is a promising but extremely challenging setting where most current works are not applicable in this setting, which limits the generalizability of these works. Cradle manages to **show the effectiveness of GCC setting**, which we hope can attract more researchers to engage in this exciting domain to achieve the general computer agents. And Cradle also serves as the **first open-sourced framework** for the researchers to start with. None of the above-mentioned will be possible without your support.
> > > > > > >
> > > > > > > We thank the reviewers once again for your efforts and hope reviewers can re-evaluate our paper.
> > > > > > >
> > > > > > > Best regards,
> > > > > > > Authors of Paper 6747

---

> ### Author Response · Authors · 2024-11-17
>
> **Q7**: The baseline comparisons in section 4.1. do not seem all that useful/informative to me. It is not clear what is being 'ablated' compared to Cradle (GPT-4o) in Table 3.
>
> **A7**: Sorry for the confusion. There is a mistake in the caption of Table 3, where the ablation studies should be baseline comparisons. Table 3 is also only mentioned in the Baseline Comparison Section. We have fixed it in the latest version.
>
> ---
>
>
> **Q8**: The ablations in 4.2 could be expanded to at least one non-game scenario tasks. The authors present the system as one that generalizes across multiple domains and it would be useful for readers to know what elements of the system are most helpful for enabling that generalization or indeed if non-video game domains need the same set of components as video game domains. I'd encourage the authors to run the ablations over the OSWorld benchmark or a benchmark like WebArena (which has wider adoption) that enables automatic evaluation.
>
> **A8**: Thanks for the suggestion. Before the deadline of the submission, OSWorld is the only benchmark that satisfies GCC setting, which directly uses keyboard and mouse actions to interact with. Other benchmarks like WebArena only provide the interactable elements ID obtained from the internal API for the agents to interact with, which violates the GCC setting.
>
> Running experiments on such a comprehensive benchmark is time- and budget-consuming. As mentioned in Table 6, a single run on OSWorld would cost about 500 USD and 240 hours. We hope the reviewers can understand that it is too costly to conduct ablation studies on such a benchmark.
>
> ---
>
> **Q9**: Another ablation that I think would make this a stronger paper is a version of Cradle where the LLM doesn't have access to tools such as Grounding DINO, SAM and others. Is that they key ingredient to make systems like this effective.
>
> **A9**: Thanks for pointing it out. We would like to mention that these tools are integrated within the Information Gathering module. When the module is ablated in the ablation studies, these tools are also unavailable. Only RDR2 uses Grounding DINO to help with enemy detection, where the performance will drop from 20% to 0 if without the tool in the combat task. These tools are only used to solve the rare cases that LMM can not handle. In most cases, Cradle still relies on the LMMs’ own visual abilities. Other games like Stardew Valley, Cities: Skylines and Dealer’s Life 2 do not rely on these tools.
>
> ---
>
> **Q10**: Table 1: What are the units for "Farm Clearup (Grids Num)" and "Avg Haggling (Count)"?
>
> **A10**: The unit is exactly shown in the brackets, namely the number of grids that the agent clears up and the average count of haggling per deal.
>
> ---
>
> **Q11**: When you use the plus or minus sign, is that reflecting the range in both directions? Cities Skylines Human population is 415+-416, which would imply some player had -1 population? This table might be clearer if it indicated the min and max separately.
>
> **A11**: 415±416 indicates that the mean of the population is 415 with a standard deviation of 416, which is the standard and most widely accepted expression. The population should be at least 0 without an upper limit.
>
> ---
>
> **Q12**: Table 4: I think this should be be "Figure" 4. I do find it hard to read, what do the shaded areas represent? They seem to overlap a fair amount, how should we interpret this?
>
> **A12**:  It is the error bar, which can be found in almost every RL and other decision-making paper. It represents the standard deviation of the line with the same color. The overlaps are quite common in this kind of figure, which is usually caused by two lines with close value and std. They do not have any special meaning.

---

> > ### Author Response · Authors · 2024-11-17
> >
> > **Q15**: Are the authors proposing the GCC (General Computer Control) setting as a contribution/novel aspect of this work? Could the authors clarify how the proposed GCC differs from the general approach to learning control from pixels Human-level control through deep reinforcement learning [Mnih et al 2016]?
> >
> > **A15**: Yes. GCC setting is our primary contribution. Please refer to Q1 1.1.
> >
> > Traditional DRL usually works on open-sourced environments, like Atari and MuJoCo, where the reward function relies on access to the internal state of the environments. In close-sourced environments, it is difficult to design rewards for agents to learn. Also, different environments have different observation and action spaces,  agents trained in one environment cannot be deployed to another environment with different obs and action spaces. It is the GCC setting that guarantees the same obs and action spaces across all the environments and Cradle does not rely on explicit reward to improve.
> >
> > ---
> >
> > **Q16**: How is a 'step' defined in the context of the games/cradle? Is one step=one action selected by cradle? How many mouse/keyboard interactions can happen in one step?
> >
> > **A16**: Yes. One step is one action selected by Cradle. An action is usually made up of 1-3 skills. Most skills contain 1-3 keyboard and mouse interactions and a few skills can have more than 3 interactions.
> >
> > ---
> >
> >
> > **Q17**: In table 1 are the humans limited in the amount of time they get to spend to complete the tasks?
> >
> > **A17**: The limit is 2 hours but no one exceeds the limit.
> >
> > ---
> >
> > **Q18**: How customized do the prompts for each module in Cradle need to be for each domain it is applied to? Is there anything that can be shared across multiple domain without heavy customization?
> >
> > **A18**: For every domain, the prompts share the same template in each module, which contains the minimal requirements of the fields and keys of input and output of the module. These templates can be directly run, however, without any performance guaranteed. Users can freely add domain knowledge in the corresponding field in the template or change the input and output as they wish. The level of customization fully depends on the difficulty of the domain and how the LMM is familiar with the domain. For example, in OSWorld, Cradle uses the same prompt to deal with dozens of software across hundreds of tasks. For each game, Cradle needs to apply customized prompts for the challenging tasks.
> >
> > ---
> >
> > **Q19**: What happens to visual information/screenshots during the summarization process into episodic memory?
> >
> > **A19**: They will be summarized into text and stored in the episodic memory.
> >
> > ---
> >
> > **Q20**: In appendix H.4 how is "visual prompting" different from the information gathering used in other scenarios? If the augmented screenshot is not fed to the information gathering modules, where does it go?
> >
> > **A20**: In other modules like Self-Reflection, Task Inference and Action Planning, the screenshots will also serve as part of the input. If the "visual prompting" is applied, the input will be replaced with augmented screenshots.

---

### Official Review · Reviewer_w2hJ · 2024-11-05

**Soundness:** 3
**Presentation:** 2
**Contribution:** 2
**Rating:** 5
**Confidence:** 4

**Summary:**

The paper proposes General Computer Control (GCC) setting in a pursuit to build foundation agents that can master ANY computer task via the universal human-style interface by receiving input from screens and audio and outputting keyboard and mouse actions. The solution approach backed by LMM is broken into 6 modules - 1) information gathering 2) self-reflection 3) task inference 4) skill curation 5) action planning. and 6) memory. Each of the module is challenging and having it's own pros & cons and it is good to see a pipeline stitched in the paper, whose efficacy is further demonstrated in different categories of tasks. Overall, this is a good systems paper with limited novelty and contribution in the representation learning space.

**Strengths:**

The system architecture is well thought of. It uses LMMs to generate code functions as semantic-level skills, which encapsulate lower-level keyboard and mouse control.

Codebase has been shared and the appendix is extensive with figures and supporting material, case studies. Hard work has been put, but mostly on the engineering side.

The usage in software tasks is something fresh, although tech is a join of off the shelf SOTA.

Formulation / design of the template prompts is good with example and format. This seems to be in line with Robogen [https://github.com/Genesis-Embodied-AI/RoboGen] philosophy.

**Weaknesses:**

To establish the approach firmly, it is recommended that the authors start the paper with clear input & output example. Fig 15 comes later in Appendix, but is cumbersome to fit in front. The system architecture at front part of paper is too much information right at front, without diving in the actual technical challenges. I think a higher level pipeline [input, output, objective] should have been shown and then specific details on sub-objectives and I/O.

Episodic Memory - there is a lot of important recent work, specifically in egocentric video - FPS. What novelty is bought here is not clear as this is one of the driving factors of success of downstream task [in lieu of earlier skill learning].

Reasoning is also handled by LMM as it is, thereby giving credit to the LMM paper, instead of this paper. At least some sort of improvement LMM++ was expected.

Not sure if the approach is generalization to different domains and how much depends on perception part? More errors there -> more errors downstream.

**Questions:**

In terms of action planning, whether PDDLstream could have been explored?

If the pipeline so heavily depends on LMM, then if LMM become very superb in near future as encompassing the architecture internally, then the pipeline might not be needed and go towards dissolution.

How would have Deep RL fared under this scenario with game related rewards?

Is there any LMM query prompting level innovation like chain of thought incorporation

The metrics of evaluation vary per dataset / game / software category. How can this be generalized to reduce human input?

What mathematics and philosophy is behind the fact that this chain of 6 modules will work perfectly, and not any other combination?

Were there any robust human user studies done domain and demography nostic?

I was wondering how will it work for CAPTCHA task?

**Details Of Ethics Concerns:**

The paper, webpage and product offering is available on the internet thereby revealing author and organization details.

---

> ### Author Response · Authors · 2024-11-17
>
> We thank the reviewers for their valuable feedback and insightful comments.  We hope our following answers will clear up the doubts about our work, and please let us know if there is any other clarification we can provide.
>
> ---
>
> **Q1**: Overall, this is a good systems paper with limited novelty and contribution in the representation learning space.
>
> **A1**: We gratefully appreciate the reviewer’s acknowledgment of our paper on the system/application side. We want to argue that our novelty and contribution are still within the scope of representation learning, which is actually a large and general concept in machine learning including both proposing new methods for learning better representation and also the application of representation learning method.
>
> First of all, according to according to ICLR2025 Call for Papers (https://iclr.cc/Conferences/2025/CallForPapers), "We consider a broad range of subject areas including feature learning, metric learning, compositional modeling, structured prediction, reinforcement learning, uncertainty quantification and issues regarding large-scale learning and non-convex optimization, **as well as applications** in vision, audio, speech, language, music, robotics, games, healthcare, biology, sustainability, economics, ethical considerations in ML, and others."
>
> Representation learning is just one subject area in the listed 20 subject areas. Other subject areas include "datasets and benchmarks", "infrastructure, software libraries, hardware, etc" and applications to various domains. The novelty of papers in these subject areas usually does not come from proposing a novel method to learn better representation.
>
> Our selected area of this paper is **applications** to robotics, autonomy, **planning**, focusing on the application of representation learning. As a reference, MetaGPT[1], accepted by ICLR2024 as oral presentation, which proposes a code-generation framework for LLM-based multi-agent collaborations without training, was also submitted to this primary area. They also focus on building LLM "systems" instead of "representation learning", which is exactly the same as our paper. Another famous paper, ReAct[2], accepted by ICLR2023 as notable-top-5%, which is one of the most popular agent frameworks, was also submitted to the area of applications.
>
> We hope the reviewer can re-evaluate the novelty and contribution of our paper based on the above information since the reviewer has already acknowledged the novelty and contribution of our paper required by our selected area.
>
> [1] Hong, Sirui, et al. "Metagpt: Meta programming for multi-agent collaborative framework." ICLR 2024.
> [2] Yao, Shunyu, et al. "ReAct: Synergizing Reasoning and Acting in Language Models." ICLR 2023.
>
> ---
>
> **Q2**: The authors should start the paper with clear input & output examples, where Fig 15 in the appendix is cumbersome to fit in front. A higher level pipeline [input, output, objective] should have been shown and then specific details on sub-objectives and I/O.
>
> **A2**: We agree that the paper should start with the input & output example, which is exactly how we organize the paper. At the beginning of the framework section, we use Figure 3 to show the input and output of each module to provide a high-level understanding to the readers. Then in Figure 5, we provide 3 complete examples in different games to show how Cradle works during each step with real input and output examples. We appreciate the reviewers’ thorough reading of the appendix and their insightful observations regarding Figure 15. However, Figure 15 does not show the complete workflow of each module, which is used as a case study of how the reflection module works. We believe Figure 3 and 5 are sufficient to demonstrate the input & output and provide a precise understanding of our framework.
>
> As for the technical challenges and specific details, we argue that Cradle is a general framework rather than a specific method, where each module can have different env-specific implementations to deal with env-specific issues. As a general framework, Cradle does not aim to focus on one or two software and their corresponding implementations. The framework section in the main paper should focus on providing a clear explanation of the framework instead of any specific implementations. And we also agree that the technical details are very important, so we tried our best to provide all the details we can provide in the appendix for the readers.

---

> > ### Author Response · Authors · 2024-11-17
> >
> > **Q3**: Episodic Memory - there is a lot of important recent work, specifically in egocentric video - FPS. What novelty is bought here is not clear as this is one of the driving factors of success of downstream task [in lieu of earlier skill learning].
> >
> > **A3**: We want to clarify again that Cradle is a general and flexible framework rather than a specific method, each module can have different task/env-specific implementations. The work mentioned by the reviewer does not have any competitive or substitutive relationship with Cradle. Instead, they can be integrated into Cradle as part of the module to improve the overall performance of the framework. And the skill learning process happens in the skill curation module instead of episodic memory. Our novelty comes from the whole framework rather than the specific implementation of the module.
> >
> > ---
> >
> > **Q4**: Reasoning is also handled by LMM as it is, thereby giving credit to the LMM paper, instead of this paper. At least some sort of improvement LMM++ was expected.
> >
> > **A4**:  Though it is not clear to us what exactly LMM++ is, we want to emphasize that the improvement of reasoning ability does not have to come from the improvement of LMM itself. Even the COT paper does not do any modification to the LLM. Tons of papers, particularly in the agent domain, study how to establish reasonable pipelines to unlock the potential reasoning ability of LLMs without any finetuning, such as ReAct[1], Reflextion[2], Voyager[3], CAMEL[4], MetaGPT[5], AutoGen[6] and Mobile-Agent-v2[7] and so on. Cradle significantly outperforms baseline methods across all tasks, already showing its superior reasoning ability, whereas only GPT-4o alone struggles with completing any of them.
> >
> > [1] Yao, Shunyu, et al. "ReAct: Synergizing Reasoning and Acting in Language Models." ICLR 2023.
> > [2] Shinn, Noah, et al. "Reflexion: Language agents with verbal reinforcement learning." NeurIPS 2023.
> > [3] Wang, Guanzhi, et al. "Voyager: An Open-Ended Embodied Agent with Large Language Models." Transactions on Machine Learning Research.
> > [4] Li, Guohao, et al. "Camel: Communicative agents for" mind" exploration of large language model society." NeurIPS 2023.
> > [5] Hong, Sirui, et al. "Metagpt: Meta programming for multi-agent collaborative framework." ICLR 2024.
> > [6] Wu, Qingyun, et al. "Autogen: Enabling next-gen llm applications via multi-agent conversation framework." COLM 2024.
> > [7] Wang, Junyang, et al. "Mobile-Agent-v2: Mobile Device Operation Assistant with Effective Navigation via Multi-Agent Collaboration." NeurIPS 2024.
> >
> > ---
> >
> >
> > **Q5**: Not sure if the approach is generalization to different domains and how much depends on perception part? More errors there -> more errors downstream.
> >
> > **A5**: Cradle is designed to complete computer tasks under GCC setting. The generalization comes from the GCC setting, where Cradle only takes pixels shown on the screen (screenshots) as input and controls keyboard and mouse to interact with computers. Theoretically, Cadle is able to interact with any software. Cradle is evaluated on 4 challenging video games (which have never been explored before), 5 software applications and a comprehensive benchmark, OSWorld. We believe the empirical studies are sufficient to show Cradle’s superior generalization.
> >
> > Perception has been one of the key components of agents, no matter whether they are RL agents, LMM agents, or robotics. If agents cannot understand the obs, it is difficult for them to make reasonable decisions. Compounding errors occur at every stage of the decision-making process, which is a general challenge in decision-making tasks and is not necessarily a specific weakness of our paper.
> >
> > ---
> >
> > **Q6**: In terms of action planning, whether PDDLstream could have been explored?
> >
> > **A6**: Yes, PDDLstream could have the potential to be integrated into the action planning module, as long as LLMs can write PDDL, whereas it might be challenging to describe the computer task and explicitly write the corresponding constraints with PDDL. So we mainly follow the Code as Policies[1] style in this work.
> >
> > [1] Liang, Jacky, et al. "Code as policies: Language model programs for embodied control." ICRA 2023.
> >
> > ---
> >
> > **Q7**: If the pipeline so heavily depends on LMM, then if LMM become very superb in near future as encompassing the architecture internally, the pipeline might not be needed and go towards dissolution.
> >
> > **A7**: Thanks for pointing it out. As we mentioned in the limitation and future work section,  "The six modules represent a problem-solving mindset; as LMM capabilities improve, some or even all of these modules may be combined into a single request". Cradle is the first attempt towards GCC but definitely not the ultimate answer to GCC. We hope this work can motivate more researchers to join to find better solutions. How to encompass the architecture internally is absolutely a promising direction for future research.

---

> ### Author Response · Authors · 2024-11-17
>
> **Q8**: How would have Deep RL fared under this scenario with game related rewards?
>
> **A8**: Thanks for pointing it out. This is exactly one of the challenges of GCC setting. In these closed-source environments, it is extremely difficult to design reward functions as encapsulations of the environments since all the users receive are screenshots/videos from the monitor without any internal information. There are no available easy-to-obtain reward signals for RL agents to learn.
>
> ---
>
> **Q9**: Is there any LMM query prompting level innovation like chain of thought incorporation
>
> **A9**: It is not clear to us what is the LMM query prompting level innovation. To our understanding, CoT is a method that unlocks the potential reasoning ability of LMMs. The following works like ReAct[1] and Reflextion[2] also attempt to build reasonable pipelines to increase the reasoning ability. Cradle also serves a following work of them. If the reviewers acknowledge the innovation made by ReAct and Reflextion, then they should also acknowledge the innovation made by Cradle.
>
> [1] Yao, Shunyu, et al. "ReAct: Synergizing Reasoning and Acting in Language Models." ICLR 2023.
> [2] Shinn, Noah, et al. "Reflexion: Language agents with verbal reinforcement learning." NeurIPS 2023.
>
> ---
>
> **Q10**: The metrics of evaluation vary per dataset / game / software category. How can this be generalized to reduce human input?
>
> **A10**: We would like to note that due to the rich content in games and software, it is difficult to use only one or a few metrics for them. Different games may evaluate agent’s different abilities, just like that each industry has its own metrics for measurement, e.g., sellers use the profits, runners use the time and students use the scores. For the goal-based tasks, a common metric would be the success rate. For the creative tasks, there are usually various metrics for them.
>
> ---
>
> **Q11**: What mathematics and philosophy is behind the fact that this chain of 6 modules will work perfectly, and not any other combination?
>
> **A11**: As we introduce in the section 3.3, Cradle’s design represents a problem-solving mindset, whose reasoning process is analogous to "reflect on the past, summarize the present, and plan for the future".
>
> Information Gathering and Self-Reflection are used to analyze the performance and outcomes of the preceding action's execution.
>
> Task Inference and Episodic Memory summarize the current situation and analyze the current progress of the task based on the reflection results.
>
> Skill Curation and Action Planning generate and select skills to be executed in the next actions,
> according to the current situation and the progress of the task.
>
> We will not say this framework works perfectly as if we add more auxiliary modules, the performance can still be improved, which is a trade-off between the performance and time/budget-cost. However, as shown in the ablation studies, removing any of them will result in a significant performance drop.
>
> ---
>
> **Q12**: Were there any robust human user studies done domain and demography nostic?
>
> **A12**: We are sorry that we may not fully understand the meaning of this question. It would be very helpful if the reviewers can re-clarify this question.
>
> ---
>
> **Q13**: I was wondering how will it work for CAPTCHA task?
>
> **A13**: For the CAPTCHA task, it mainly depends on the LMMs’ own ability, especially their visual ability. Due to the limited visual understanding mentioned in the paper, we do not expect the current LMMs will generally have a good performance across various CAPTCHA tasks without any specific tools.
>
> ---
>
> **Q14**: About ethics issue. The paper, webpage and product offering is available on the internet thereby revealing author and organization details.
>
> **A14**: We would like to remind the reviewers that it is not an ethics-related issue. And according to the ICLR policy, "papers that have appeared on non-peer reviewed websites (like arXiv) or that have been presented at workshops (i.e., venues that do not have publication proceedings) do not violate the policy."

---

> > ### Author Response · Authors · 2024-11-24
> >
> > We sincerely appreciate the reviewers' effort during the review process and deeply understand the reviewers' busy schedule.  We hope our responses above manage to solve all the reviewers' concerns.  As the deadline for rebuttal is approaching,  we would like to know whether there are any other concerns or questions that we can clarify. It is the reviewers' insightful feedback that makes our work much stronger.

---

> ### Author Response · Authors · 2024-11-28
>
> Dear Reviewer w2hJ,
>
> We sincerely appreciate your insightful comments and the acknowledgment of our paper from the system/application side. As the rebuttal deadline approaches, we would like to know whether there are any additional concerns we can address. We will try our best to provide further clarification.
>
> GCC is a promising but extremely challenging setting where most current works try to avoid it resulting in limited generalizability. **Cradle manages to show the effectiveness of GCC setting, which we hope can attract more researchers to engage in this exciting domain to achieve the general computer agents. And Cradle also serves as the first open-sourced framework for the researchers to start with. None of the above-mentioned will be possible without your support.**
>
> We hope our provided response during the rebuttal addresses the reviewers' concerns and the reviewers can re-evaluate our paper.  We thank the reviewers once again for their efforts.
>
>
> Best regards,
> Authors of Paper 6747

---

> ### Author Response · Authors · 2024-12-02
>
> Dear Reviewer w2hJ,
>
> We want to again express our gratitude to your efforts and insightful comments. As it is the **last day** of rebuttal for reviewers to reply, we hope our provided responses have addressed all the concerns. Our rebuttal can be briefly summarized as following three points:
>
> 1. **Application is within the acceptable scope of ICLR**, which does not necessarily require novelty in learning better representation.
>
> 2. Our framework has been described in Figures 3 and 5, which clearly shows the input and output of each module and provides a precise understanding of our framework. **Cradle is a general framework, instead of a specific method/implementation**, which cannot include too many implementation details in the framework section.  We added a “General Implementation” subsection to the Empirical Studies Section on page 6 to provide more details.
>
> 3. We did not claim contribution from any independent module, instead, **the novelty comes from the whole Cradle framework as an integration**. As a general framework, advanced methods can be seamlessly integrated into the corresponding module and therefore enhance the ability of Cradle.
>
> GCC is a promising but extremely challenging setting where most current works are not applicable in this setting, which limits the generalizability of these works. Cradle manages to **show the effectiveness of GCC setting**, which we hope can attract more researchers to engage in this exciting domain to achieve the general computer agents. And Cradle also serves as the **first open-sourced framework** for the researchers to start with. None of the above-mentioned will be possible without your support.
>
> We thank the reviewers once again for your efforts and hope reviewers can re-evaluate our paper.
>
>
> Best regards,
> Authors of Paper 6747

---

### Meta-Review · Area_Chair_5TTC · 2024-12-20

**Metareview:**

This paper details an ambitious effort to develop foundational agents for "general" computer control. As the reviewers pointed out, the framework is quite complex so the tasks it deals with. Reviewer pm9Q praised that "The system architecture is well thought of." Similarly, Reviewer bghc mentioned that "This paper presents a well-designed framework that effectively leverages LLMs to build generalized agents capable of performing tasks across diverse domains."  The reviewers also see the evaluation to be comprehensive.

That said, I feel it is difficult to grasp the fundamental breakthrough the work brings or a clear takeaway message. Reviewer pm9Q voiced the same concern by saying "I think my primary concerns are around the presentation of the work, overall I found it quite hard to extract takeaways about the design of the system..." The approach of using "a human-style interface (screen input and keyboard/mouse output)" is hardly novel as there have been a rich body of work over the past few years along the same direction. Cradle seems to be a massive integration of many existing components for sub tasks such as information gathering, self-reflection, task inference, skill curation, action planning, and memory. I feel each of these components is unsolved in their own right, as we have seen these papers in the past.

The reviewers generally agree that the paper presents a well-designed system with strong empirical results across a variety of tasks, showcasing the potential of LLM-based agents in the GCC setting. However, they also raise concerns about the novelty of the individual modules and the clarity of the presentation, particularly regarding the implementation details and the generalization ability of the framework. The authors addressed some of the concerns by clarifying the scope of their contributions, emphasizing the novelty of the GCC setting and the Cradle framework as a whole, and providing more details on the implementation and generalization aspects. They also acknowledged the limitations of their work and suggested future directions.

Overall, AC thinks the paper touches on many critical problems in the space of GCC. Yet, the work remains at the level of a coarse integration and lacks crisp research takeaways that can benefit following researchers to further explore the topic,

**Additional Comments On Reviewer Discussion:**

The author-reviewer discussion centered around the novelty of the GCC setting and the Cradle framework, the clarity of the presentation, and the generalizability of the approach. The authors emphasized the novelty of the GCC setting, which allows agents to interact with any software without relying on APIs, enabling greater generalizability. They positioned Cradle as the first open-sourced framework for GCC, demonstrating its feasibility and potential. Given the complexity of the framework, it is doubtful how easily it can be picked up by others. The authors made progress in improving clarity of the paper by adding new materials in the revision, and acknowledged their limitation regarding generalizability. Overall, there is still a lack of strong enthusiasm among reviewers even though two of them gave high scores.

---

### Decision · Program_Chairs · 2025-01-22

Reject